# Eligibility for Photoimmunotherapy in Patients with Unresectable Advanced or Recurrent Head and Neck Cancer and Changes before and after Systemic Therapy

**DOI:** 10.3390/cancers15153795

**Published:** 2023-07-26

**Authors:** Takeshi Shinozaki, Kazuto Matsuura, Wataru Okano, Toshifumi Tomioka, Yukio Nishiya, Michiko Machida, Ryuichi Hayashi

**Affiliations:** 1Department of Head and Neck Surgery, National Cancer Center Hospital East, Kashiwa 277-8577, Japan; kmatsuur@east.ncc.go.jp (K.M.); rhayashi@east.ncc.go.jp (R.H.); 2Medical Science & Operations Division, Rakuten Medical K.K., Tokyo 158-0094, Japan; michiko.machida@rakuten-med.com

**Keywords:** photoimmunotherapy, advanced head and neck cancer, timing of photoimmunotherapy

## Abstract

**Simple Summary:**

Photoimmunotherapy is a novel cancer treatment that recently became covered by national health insurance only in Japan. We investigated the characteristics of patients with head and neck cancer who can, potentially can, and cannot be treated with photoimmunotherapy (eligible, potentially eligible, and ineligible, respectively). We retrospectively reviewed the medical records of 246 patients who started receiving systemic therapy for advanced or recurrent head and neck cancer. After exclusions, 194 patients were evaluated, of whom 108 were ineligible for photoimmunotherapy. Eight patients were potentially eligible but ultimately not suitable candidates for photoimmunotherapy. The remaining nine patients were considered eligible. Of the nine eligible patients who received first-line systemic therapy, four became ineligible for photoimmunotherapy due to disease progression. Appropriately selected patients with advanced head and neck cancer are candidates for photoimmunotherapy, and this study contributes to deciding the strategy and timing of treatment for these patients.

**Abstract:**

Photoimmunotherapy is a novel cancer treatment that recently became covered by national health insurance in Japan, but treatment decision-making remains challenging for unresectable advanced or recurrent head and neck cancer. We aimed to clarify the characteristics of patients for whom photoimmunotherapy was indicated by a retrospective chart review. Patients aged ≥20 years diagnosed with advanced or recurrent head and neck cancer who started receiving systemic therapy at the National Cancer Center Hospital East from January 2016 through December 2020 were retrospectively analyzed. Before and after first-line systemic therapy, patients were classified into 3 groups according to eligibility for photoimmunotherapy: eligible, potentially eligible, and ineligible. In total, of 246 patients evaluated—194 after exclusions were analyzed—108 were deemed ineligible for treatment. Of the remaining 86 patients, 8 were considered potentially eligible and 9 eligible. Of the nine eligible patients, four became ineligible after receiving first-line systemic therapy due to disease progression. Our results suggest that the indication of photoimmunotherapy should be considered before, during, and after systemic therapy for unresectable locally advanced or recurrent head and neck cancer.

## 1. Introduction

Cetuximab is accepted worldwide in various indications for the therapy of head and neck squamous cell carcinoma (HNSCC). Photoimmunotherapy utilizes an antibody–drug conjugate, cetuximab sarotalocan sodium, that consists of cetuximab, a chimeric anti-human epidermal growth factor receptor (EGFR) monoclonal antibody (IgG1), and a light-sensitive compound, the dye IRDye^®^ 700DX (IR700) [1]. The conjugate is highly selective for binding cells expressing EGFR, which is highly and strongly expressed in HNSCC. The dye is activated by illumination of a 690 nm (red) laser beam from the BioBlade^®^ laser system and then rapidly kills only the cells to which the conjugate is bound [2,3]. This technology was developed from the Alluminox^TM^ platform [4]. Figure 1 shows the structure of the drug and an overview of photoimmunotherapy for HNSCC [2,3]. The mechanism of action is thought to be as follows: (i) activation of the antibody conjugate by laser illumination, (ii) which causes damage on the cell membrane, (iii) resulting in an increase in transmembrane water flux and (iv) leading to cell rupture and necrosis. This reaction is considered to occur within an extremely short period of time after laser illumination [3]. The rapid release of immunogenic signals from damaged cancer cells may induce maturation of dendritic cells and trigger a host immune response against the tumor based on pre-clinical (in vivo) data [5], but this has not been clinically confirmed at present.

Photoimmunotherapy for head and neck cancer (HN-photoimmunotherapy) has been investigated in a phase I/IIa clinical trial (RM-1929-101 study) conducted in the United States in 2015 [6] and in a phase I clinical trial (RM-1929-102 study) conducted in Japan in 2018 [7]. Based on the results, national health insurance coverage of HN-photoimmunotherapy in Japan began on January 1, 2021. The approved indications are unresectable locally advanced or locally recurrent head and neck cancer (HNC), including non-squamous cell carcinoma (non-SqCC). Standard therapies for HNC, such as chemoradiotherapy, are preferred over this treatment when available. However, if the disease subsequently progresses or recurs, local boron neutron capture therapy, HN-photoimmunotherapy, or systemic therapy may be selected. 

The level of EGFR expression is known to be higher in the tumor than in the adjacent normal tissue [8], but the normal tissue around the tumor may also express EGFR. Fatal complications can occur if the necrosis extends to major blood vessels such as the carotid arteries and vessels in the skull base. Data on the clinical safety of HN-photoimmunotherapy are limited [6,7,9] because the first global phase 3 trial of this treatment is currently enrolling patients (ClinicalTrials.gov identifier: NCT03769506). Although there are several case reports in the literature [10,11,12,13,14,15,16,17], no large-scale case series or long-term follow-up results have been reported for HN-photoimmunotherapy. Therefore, in clinical practice, it is often difficult to decide which treatment should be selected for unresectable advanced or recurrent disease.

Currently, no data on patient characteristics are available from medical databases for identifying the appropriate patient population for HN-photoimmunotherapy. This treatment requires an approach to the target lesion with a laser device and should be selected for patients in whom the extent of tumor necrosis will not affect the carotid artery or other vital organs. 

Thus, with the aim of improving personalized treatment for patients with HNC, in this retrospective study we sought to identify the appropriate patient population for HN-photoimmunotherapy using medical records and imaging data of patients with unresectable advanced or recurrent HNC. Furthermore, considering the difficulty of choosing between systemic therapy and HN-photoimmunotherapy in these patients, we examined changes in eligibility for HN-photoimmunotherapy based on the response to systemic therapy (individually after systemic therapy?).

## 2. Materials and Methods

Medical records were retrospectively reviewed for patients aged ≥ 20 years diagnosed with advanced or recurrent HNC who started receiving systemic therapy at National Cancer Center Hospital East from January 2016 through December 2020. A physician with experience in at least 10 cases of HN-photoimmunotherapy independently judged eligibility for the treatment. We examined data including imaging just before first-line systemic therapy. We excluded patients with non-SqCC and those without imaging of the recurrent lesion because it was not indicated for HN-photoimmunotherapy. Table 1 shows the evaluated items. Cases were classified into 3 groups according to eligibility for HN-photoimmunotherapy: eligible, potentially eligible, and ineligible. Patients with a tumor close to the carotid artery or skull base, those with bone invasion, and those who could not tolerate general anesthesia were deemed ineligible for HN-photoimmunotherapy because they are considered high risk based on treatment experience to date. Patients were also deemed ineligible if HN-photoimmunotherapy was not technically feasible (e.g., due to difficulty in illumination of the tumor or damage to surrounding tissue caused by the needle catheter). Patients were classified as potentially eligible if treatment was technically feasible and would not result in damage to major blood vessels or vital organs but a major reduction in quality of life was expected, such as pharyngocutaneous fistula or major facial defects.

In an exploratory analysis, we examined eligibility for HN-photoimmunotherapy before and after first-line systemic therapy to determine whether there was any change. The same method as described above was used to classify the patients as eligible, potentially eligible, or ineligible, with the additional criteria that patients with a complete response (CR) were considered to no longer need HN-photoimmunotherapy and those with highly progressive disease (PD) and thus incapable of undergoing HN-photoimmunotherapy were considered ineligible. Eligibility for HN-photoimmunotherapy before systemic therapy was examined as the primary outcome, and the change in eligibility was assessed as an exploratory outcome for this retrospective study (Figure 2).

This study was approved by the Research Ethics Review Committee of the National Cancer Center Hospital East (2021-330). As this was a noninterventional observational study, the need for informed consent was waived. 

## 3. Results

We evaluated 246 patients who initiated systemic therapy for advanced or recurrent HNC at the National Cancer Center Hospital East from January 2016 to December 2020. Of these, 52 cases were excluded for the following reasons: non-SqCC, 47 patients; started palliative systemic therapy at another hospital, 2 patients; undifferentiated thyroid cancer, 2 patients; clinical trial participation, maintenance chemotherapy after chemoradiotherapy, and not treated at our hospital, 1 patient each. The background characteristics of the 194 patients are shown in Table 2, and their history of previous treatment is shown in Table 3. Eleven patients had no prior history of surgical resection or radiotherapy, and most of them had distant metastases. 

### 3.1. Eligibility for HN-Photoimmunotherapy

Of the 194 patients evaluated in this study, 69 had no lesions in the head and neck region. The remaining 125 patients were reviewed to determine their eligibility for HN-photoimmunotherapy (Figure 3).

In total, 76 patients were found to be ineligible for HN-photoimmunotherapy due to the involvement or proximity of the carotid artery, 23 due to the proximity of the skull base, and 9 for other reasons (Table 4). Of these nine patients, 3 had bone invasion, one was in poor general condition and could not tolerate general anesthesia, and five had tumors for which laser illumination would be difficult (3 tumors were located in the hypopharynx or deep in the body, where laser illumination is technically challenging, and 2 were large tumors that posed high risk of mortality or reduced quality of life due to tissue loss following necrosis induced by HN-photoimmunotherapy). Of the remaining 17 patients, 8 were considered potentially eligible for HN-photoimmunotherapy and 9 were considered eligible.

The eight patients considered potentially eligible for HN-photoimmunotherapy were deemed unsuitable for the following reasons: Three were expected to have substantial loss of quality of life after treatment, and five had distant metastases to the lungs, which would dictate the prognosis. 

The nine cases considered eligible for HN-photoimmunotherapy are listed in Table 5. All but one had SqCC and had been previously treated with radiotherapy. Among the eligible patients, the root of the tongue was the most common target site (4 patients), and one patient participated in a clinical trial of HN-photoimmunotherapy.

### 3.2. Changes in Eligibility for HN-Photoimmunotherapy as a Result of Systemic Therapy

Table 6 shows the changes in eligibility for HN-photoimmunotherapy as a result of first-line systemic therapy. Of 108 patients who were ineligible for HN-photoimmunotherapy before systemic therapy, 9 (6.6%) achieved CR after systemic therapy and did not require HN-photoimmunotherapy, 3 (2.2%) became eligible for HN-photoimmunotherapy due to partial response (PR), and the remaining 96 patients remained ineligible. Of the 17 patients who were considered eligible or potentially eligible before systemic therapy, none of the 8 potentially eligible patients became eligible after systemic therapy. Of the nine eligible patients, four became ineligible due to PD, two remained eligible while they maintained a PR by systemic therapy despite their recurrent lesions in the root of the tongue, and three could not be evaluated because they participated in other clinical trials without receiving the planned first-line systemic therapy. One eligible patient turned out to be ineligible because general anesthesia was not possible due to an immune-related adverse event induced by systemic therapy.

### 3.3. Cases That Changed from Ineligible to Eligible for HN-Photoimmunotherapy following Systemic Therapy 

We next present imaging findings from two cases that changed from ineligible to eligible for HN-photoimmunotherapy following first-line systemic therapy. 

In Case 1 (Figure 4), the patient had SqCC of the root of the tongue (cT3N0M0). Chemoradiotherapy with cisplatin was administered, but the patient developed local recurrence and cervical lymph node metastasis. This patient was deemed ineligible for HN-photoimmunotherapy before systemic therapy because the tumor invaded the larynx and posed a risk of massive necrosis following this treatment. In addition, a metastatic lymph node was in contact with major vessels. After systemic therapy with paclitaxel and cetuximab, the primary tumor shrank and the metastatic lesion disappeared. Because the downsized tumor no longer posed a risk of massive necrosis, the patient became eligible for HN-photoimmunotherapy. 

In Case 2 (Figure 5), the patient had SqCC of the root of the tongue (T4aN3M0). Induction chemotherapy and radiotherapy with cetuximab were administered, but the tumor remained. Before first-line systemic therapy, the tumor and metastatic lymph nodes were infiltrating major blood vessels, and thus the patient was ineligible for HN-photoimmunotherapy. After systemic therapy with nivolumab, the primary tumor and lymph nodes infiltrating major vessels had disappeared on contrast-enhanced computed tomography. The remaining metastatic lesions were in a right subcutaneous node and in a left cervical node that could be treated by HN-photoimmunotherapy and neck dissection, respectively.

## 4. Discussion

HN-photoimmunotherapy is a local treatment that does not burden other organs and can be repeated, making it applicable to lesions that cannot be treated with drugs or radiotherapy. Although 2 years have passed since HN-photoimmunotherapy became an insured treatment for unresectable advanced HNC, including non-SqCC, in Japan, there have been no reports on the patient selection criteria and timing of this treatment. This study excluded non-SqCC patients because they do not have EGFR status information, which is a limitation of this study. The results revealed that HN-photoimmunotherapy would be technically feasible in 8.8% (17/194) of patients with advanced or recurrent HNSCC who are scheduled to receive systemic therapy. If EGFR can be measured in non-SqCC patients, this indication rate could be further expanded. In some previous reports, HN-photoimmunotherapy was used to treat patients with advanced or recurrent HNC who had exhausted other treatment options [6,12,14]; we have seen its effectiveness at our institution [7]. In patients with relapsed metastatic HNSCC, incurable locoregional disease remains a strongly adverse prognostic factor [18], and HN-photoimmunotherapy is a promising alternative for treatment in such cases. HN-photoimmunotherapy was found to be feasible in a portion of patients with advanced or recurrent HNC, and therefore it is necessary to consider eligibility for HN-photoimmunotherapy. 

The feasibility of photoimmunotherapy is affected by anatomical factors such as proximity of the tumor to the carotid arteries or skull base; the expected treatment effect, such as damage to surrounding tissue; and technical factors such as the difficulty of illumination. These factors are related to limitations of the device itself, as well as limited experience since photoimmunotherapy was introduced. With device development and the accumulation of experience, photoimmunotherapy may become feasible in a larger proportion of patients with HNC.

In addition to technical feasibility, it is also necessary to consider quality of life. Although HN-photoimmunotherapy was judged to be technically feasible in 8.8% of the patients in this study, about half of them were not suitable for this treatment due to expected major deterioration in quality of life or to metastatic sites that dictate poor prognosis. Consideration of safety should be prioritized in these patients because the effect of this treatment on the surrounding tissues cannot be excluded.

Additionally, the timing of HN-photoimmunotherapy should be considered. This is the first study that assessed whether locally advanced or recurrent HNC patients still can receive HN-photoimmunotherapy even after systemic therapy. Our results revealed that half of the patients from the eligible population became ineligible for HN-photoimmunotherapy because of progressive disease after systemic therapy. On the other hand, the remaining two patients were still eligible only during effective systemic therapy. Thus, this indicates that HN-photoimmunotherapy should be considered as an option for treatment before systemic chemotherapy in order to not lose the appropriate timing of the treatment. However, the impact that HN-photoimmunotherapy may have on the next treatment is unknown because the data related to HN-photoimmunotherapy are very limited. As published data reported that tumor bleeding, pain, and normal tissue damage are common adverse events [6,7,9,15,17], these events caused by HN-photoimmunotherapy may eventually decrease the patients’ QOL.

This study has some limitations. First, it featured a retrospective, observational design and was conducted at a single center. Second, a comparison was not made with patients who did not receive HN-photoimmunotherapy, meaning that the efficacy and safety of this treatment cannot be determined from our results. Finally, we were not able to evaluate the cases in which the patient’s general condition, such as poor renal function, prevented systemic therapy, but photoimmunotherapy might have been a better option. 

## 5. Conclusions

We identified appropriate candidates and timing for HN-photoimmunotherapy in patients with locally advanced or recurrent head and neck cancer by a retrospective chart review. Our results suggest that this treatment should be considered before and during, and not just after, systemic therapy, and that HN-photoimmunotherapy can be a treatment option for more patients with advanced or recurrent head and neck cancer.

## Figures and Tables

**Figure 1 cancers-15-03795-f001:**
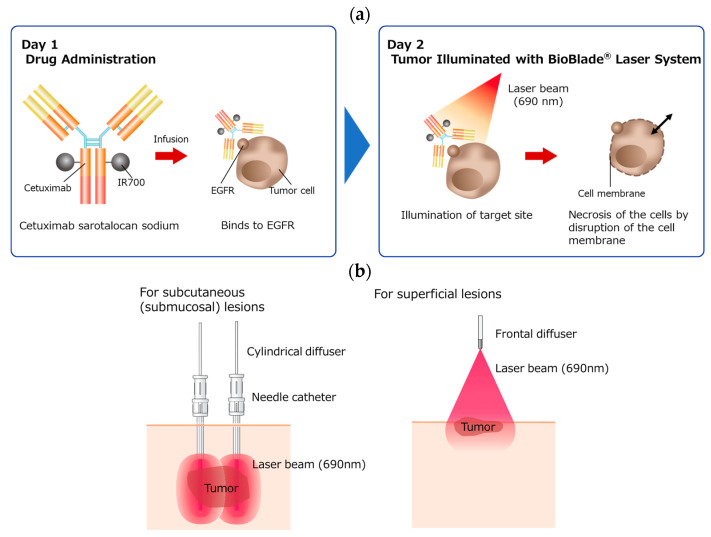
Schematic diagram of photoimmunotherapy with cetuximab sarotalocan sodium and the BioBlade laser system. (**a**) After intravenous infusion of the drug over 2 h on day 1, the tumor target is illuminated with laser light (690 nm) on day 2, leading to necrosis of the tumor by biophysical processes that damage the membrane integrity of tumor cells. (**b**) For subcutaneous (submucosal) lesions, cylindrical diffusers placed in needle catheters are used. For superficial lesions, a frontal diffuser is used.

**Figure 2 cancers-15-03795-f002:**
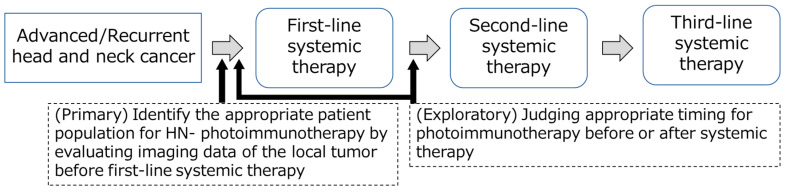
Timing of evaluations in treatment course.

**Figure 3 cancers-15-03795-f003:**
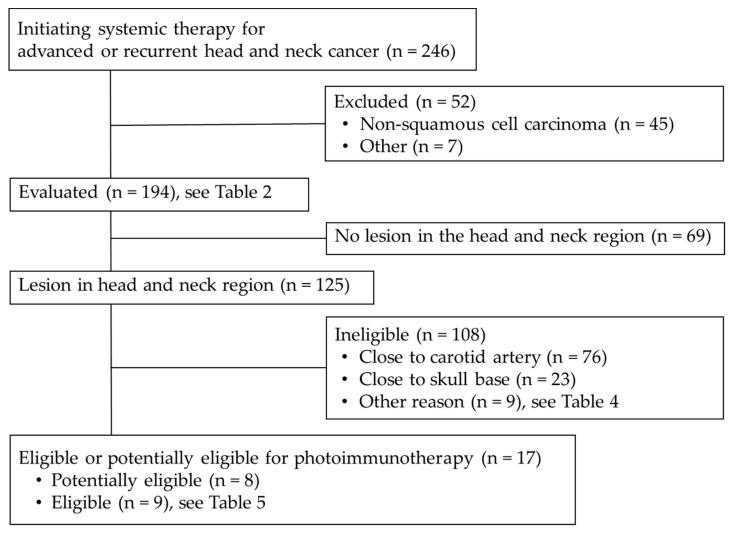
Flowchart of patients.

**Figure 4 cancers-15-03795-f004:**
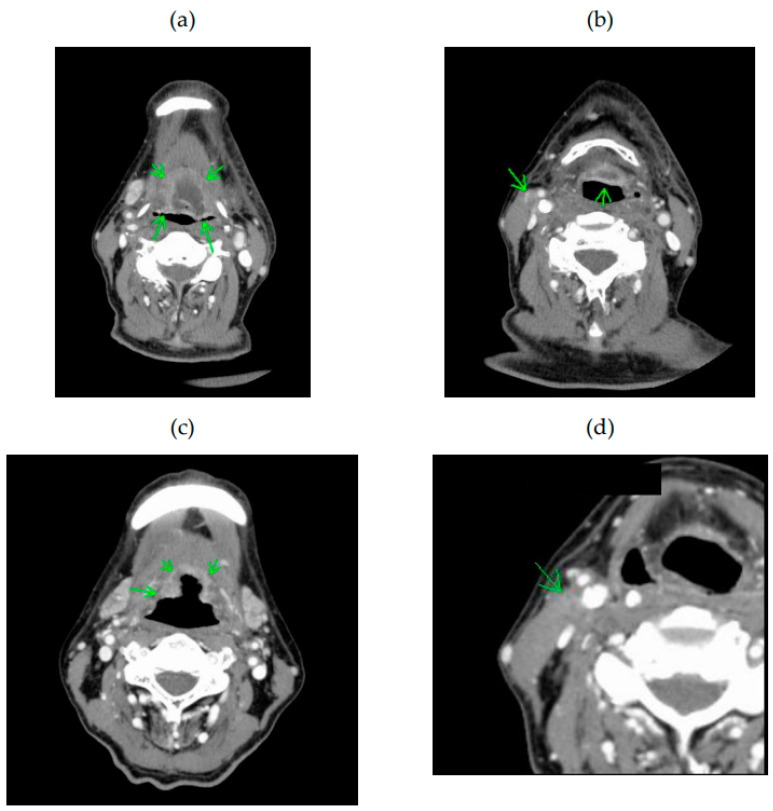
Contrast-enhanced computed tomography findings in Case 1. (**a**,**b**) Squamous cell carcinoma of the root of the tongue (arrows) and metastatic lymph node (arrow) before systemic therapy. After systemic therapy, (**c**) the tumor is smaller (arrows) and (**d**) the metastatic lymph node has disappeared (arrow).

**Figure 5 cancers-15-03795-f005:**
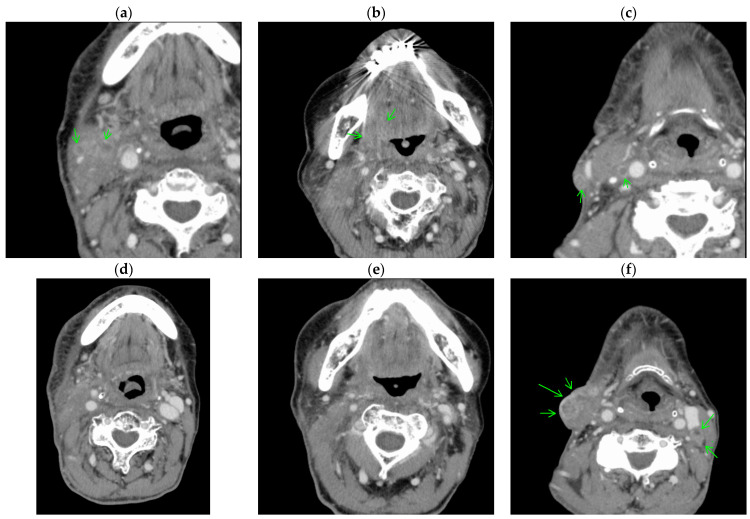
Contrast-enhanced computed tomography findings in Case 2. (**a**–**c**) Squamous cell carcinoma of the root of the tongue (T4aN3M0) (arrows) and metastatic lymph nodes (arrows) showed infiltration to the major blood vessels before systemic therapy. (**d**–**f**) After systemic therapy with nivolumab, the primary tumor and lymph nodes infiltrating major vessels have disappeared. The remaining right subcutaneous node (arrows) can be treated by HN-photoimmunotherapy and the remaining left cervical node (arrows) by neck dissection.

**Table 1 cancers-15-03795-t001:** Items evaluated in this study.

Patient Background Characteristics	HN-Photoimmunotherapy-Related Information
Age Sex Original disease Pathological diagnosis Treatment history Previous surgical resection Previous radiation exposure	Site of recurrence (lesion in the head and neck region) Tumor size (include depth) Proximity to carotid artery Bone invasion Proximity to vital organs (e.g., skull base) Impact on quality of life (e.g., risk of fistula) Feasibility of laser illumination Tolerate general anesthesia

**Table 2 cancers-15-03795-t002:** Patient demographic and clinical characteristics (n = 194).

Characteristic	n (%)
Median age (range), years	62 (20–81)
Sex (male/female)	143 (73.7%)/51 (26.3%)
Primary site	
Oral	60 (30.9%)
Nasopharynx	9 (4.6%)
Oropharynx	48 (24.7%)
Hypopharynx	43 (22.2%)
Larynx	14 (7.2%)
Nasal/paranasal	15 (7.7%)
External auditory canal	2 (1.0%)
Primary unknown HNSCC	2 (1.0%)
Salivary gland	1 (0.5%)
With distant metastasis	117 (60.3%)

HNSCC, head and neck squamous cell carcinoma.

**Table 3 cancers-15-03795-t003:** History of previous treatment (n = 194).

Previous Treatment	Radiotherapy (+)	Radiotherapy (−)	Total
Surgery (+)	97	24	121 (62.4%)
Surgery (−)	62	11	73 (37.6%)
Total	159 (82.0%)	35 (18.0%)	

**Table 4 cancers-15-03795-t004:** Cases deemed ineligible for HN-photoimmunotherapy (n = 9).

Reason for Ineligibility	Primary Site	Local Recurrent Lesion	Distant Metastasis
Bone invasion (n = 3)	Lower gingiva	Mandible	Liver
	Hard palate	Hard palate	-
	Upper gingiva	Upper gingiva	-
Poor general condition (n = 1)	Tongue	Submandibular	Bone
Difficulty in laser illumination (n = 5)	Hypopharynx	Hypopharynx	-
	Paranasal	Deep lymph node	Pleura
	Oropharynx	Paratracheal lymph node	-
	Hypopharynx	Multiple lymph node metastasis	-
	Tongue	Tongue	Lung

**Table 5 cancers-15-03795-t005:** Eligible for HN-photoimmunotherapy (n = 9).

Primary Site	Local Recurrent Lesion	Illumination Method for HN-Photoimmunotherapy (Using Device)	Target Size [mm]
Upper gingiva	Upper gingiva	Through the oral cavity (CD)	53.4 × 19.0 × 12
Tongue	Jaw	Through the cervical skin (CD)	40.4 × 24.3 × 24
Tongue	Root of tongue	From the cervical skin and oral cavity (CD)	31.1 × 20.2 × 21.0
Buccal mucosa	Skin	Through the cervical skin (CD) + surface illumination (FD)	19.8 × 15.3 × 10.9
Nasopharynx	Nasopharynx	Surface illumination through the nasal cavity (FD)	7.6 × 7.2
Hypopharynx	Cervical lymph node	Through the cervical skin (CD)	36.6 × 19.2 × 36
Hypopharynx	Root of tongue	From the cervical skin and oral cavity (CD)	8.9 × 20.7 × 15
Root of tongue	Root of tongue	Through the oral cavity (CD)	22.7 × 12.1 × 21
Root of tongue	Root of tongue	From the cervical skin and oral cavity (CD)	27.4 × 17.4 × 20.0

CD, cylindrical diffuser; FD, frontal diffuser.

**Table 6 cancers-15-03795-t006:** Changes in eligibility for HN-photoimmunotherapy as a result of systemic therapy.

Prior to Systemic Therapy	After (or While) Receiving First-Line Systemic Therapy
Ineligible (n = 108)	Remained ineligible (n = 96)
	Treatment not required after achieving CR (n = 9)
	Eligible due to PR (n = 3)
Potentially eligible (n = 8)	Ineligible due to PD (n = 8)
Eligible (n = 9)	Ineligible due to PD (n = 4)
	Remained eligible during PR (n = 2)
	NA due to participation in clinical trials (n = 3)

CR, complete response; PD, progressive disease; PR, partial response; NA, not available.

## Data Availability

The data presented in this study are available on request from the corresponding author. The data are not publicly available in order to protect the privacy of the patients.

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
