# Peer review of "Eligibility for Photoimmunotherapy in Patients with Unresectable Advanced or Recurrent Head and Neck Cancer and Changes before and after Systemic Therapy"

_cancers, 2023, doi:10.3390/cancers15153795_

Round 1
Reviewer 1 Report
An exciting analysis of the applicability of a relatively new procedure for the oncological treatment of HNSCC! The publication of these data is essential!
Nevertheless, the manuscript should be improved on some points before publication:
In general: please use abbreviations that simplify the readability of the text, e.g., instead of "head and neck carcinoma," introduce and use "HNSCC".
Introduction: The structure of the introduction is not easy to understand: please revise. E.g., start with "Cetuximab" first and its indication for the therapy of HNSCC, then the coupling of antibody with dye, then the activation of dye with laser, then the use in HNSCC.
Line 47:
"and cetuximab is used clinically to treat head and neck cancer only in Japan."
This statement is wrong -> please correct it. Cetuximab is accepted worldwide in various indications for the therapy of HNSCC.
Line 50: please simplify:
"This technology, which is and is being developed on the AlluminoxTM platform, is a drug-device combination treatment consisting of an antibody–drug conjugate and a laser device system [4]."
"This technology, which is and is being developed on the AlluminoxTM platform, is a drug-device combination treatment consisting of an antibody–drug conjugate and a laser device system 52 [4]."
Line 54:
Please specify: what is "physical stress"? Heat? Or something else?
Line 71:
"Photoimmunotherapy is highly selective for killing EGFR-positive cells, causing cell necrosis in a short time. EGFR is commonly expressed in SCC of the head and neck." -> This has already been described in the sentences above, so elaborate more in detail above or delete it.
Line 130:
“…so advanced that HN-photoimmunotherapy is not possible were considered…” -> please change to: “…so advanced that HN-photoimmunotherapy was not possible were considered…2
Line 265:
"Thus, this indicates that HN-photoimmunotherapy should be prioritized over the systemic chemotherapy in order not to lose an appropriate timing of the treatment."
-> This is a very optimistic statement, as only 6.5% of patients were initially eligible for HN photoimmunotherapy! This means that over 90% of patients needed primary palliative systemic therapy, which led to complete remission in more patients than were initially eligible for HN photoimmunotherapy.
Author Response
[Reviewer 1]
Comments and Suggestions for Authors
An exciting analysis of the applicability of a relatively new procedure for the oncological treatment of HNSCC! The publication of these data is essential!
Nevertheless, the manuscript should be improved on some points before publication:
In general: please use abbreviations that simplify the readability of the text, e.g., instead of "head and neck carcinoma," introduce and use "HNSCC".
We have used two different abbreviations, HNSCC and HNC.
HNSCC is used that it indicates squamous cell carcinoma of the head and neck, and HNC is used for indicating head and neck cancer including non-squamous cell carcinoma. (Throughout the text)
Introduction: The structure of the introduction is not easy to understand: please revise. E.g., start with "Cetuximab" first and its indication for the therapy of HNSCC, then the coupling of antibody with dye, then the activation of dye with laser, then the use in HNSCC.
We have revised the structure of the Introduction according to your suggestion. (Line 43-47)
Line 47:
"and cetuximab is used clinically to treat head and neck cancer only in Japan."
This statement is wrong -> please correct it. Cetuximab is accepted worldwide in various indications for the therapy of HNSCC.
We have revised this sentence according to your suggestion. (Line 43)
Line 50: please simplify:
"This technology, which is and is being developed on the AlluminoxTM platform, is a drug-device combination treatment consisting of an antibody–drug conjugate and a laser device system [4]."
"This technology, which is and is being developed on the AlluminoxTM platform, is a drug-device combination treatment consisting of an antibody–drug conjugate and a laser device system 52 [4]."
We have simplified the text of your point. (Line 50-51)
Line 54:
Please specify: what is "physical stress"? Heat? Or something else?
Physical stress was used to mean cell death by physical characteristics of reacted molecules. Photoimmunotherapy is indicated to cause physical changes in the shape of the antibody-IR700 conjugate due to activation by laser beam illumination after the antibody-IR700 conjugate binds to target, which induces physical stress in the cell membrane, leading to increased transmembrane water flow that eventually lead to cell bursting and necrotic cell death. (Sato K, et al. ACS Cent Sci. 2018;4:1559-1569.)
As you pointed out, it is difficult to understand, so we have revised the term to simply damage. (Line 54)
Line 71:
"Photoimmunotherapy is highly selective for killing EGFR-positive cells, causing cell necrosis in a short time. EGFR is commonly expressed in SCC of the head and neck." -> This has already been described in the sentences above, so elaborate more in detail above or delete it.
We have confirmed the repetitive phrase throughout the Introduction and corrected the points raised.
Line 130:
“…so advanced that HN-photoimmunotherapy is not possible were considered…” -> please change to: “…so advanced that HN-photoimmunotherapy was not possible were considered…2
We have revised your point. (Line 127)
Line 265:
"Thus, this indicates that HN-photoimmunotherapy should be prioritized over the systemic chemotherapy in order not to lose an appropriate timing of the treatment."
-> This is a very optimistic statement, as only 6.5% of patients were initially eligible for HN photoimmunotherapy! This means that over 90% of patients needed primary palliative systemic therapy, which led to complete remission in more patients than were initially eligible for HN photoimmunotherapy.
As you indicated, we understood the wording to be too strong for a proposal from this retrospective study. We meant that the option of HN-photoimmunotherapy should be consciously considered for treatment of locally advanced or recurrent head and neck cancer not only after systemic therapy but also before, avoid losing the opportunity for this therapy. Therefore, we have revised the wording as follows. (Line 256-258)
“Thus this indicates that HN-photoimmunotherapy should be considered as an option of the treatment before systemic chemotherapy in order not to lose an appropriate timing of the treatment.”
Reviewer 2 Report
Shinozaki et.al. investigated the characteristics of patients eligible to photoimmunotherapy for head and neck cancer. They retrospectively reviewed the medical records of patients and examined if photoimmunotherapy is possible before or after first-line systemic therapy. They conclude that photoimmunotherapy may be a treatment option for patients with advanced or recurrent head and neck cancer. The work is retrospective and preliminary but a great start to identify patients and learn about the timing of the photoimmunotherapy treatment.
Clear to understand.
Author Response
[Reviewer 2]
Shinozaki et.al. investigated the characteristics of patients eligible to photoimmunotherapy for head and neck cancer. They retrospectively reviewed the medical records of patients and examined if photoimmunotherapy is possible before or after first-line systemic therapy. They conclude that photoimmunotherapy may be a treatment option for patients with advanced or recurrent head and neck cancer. The work is retrospective and preliminary but a great start to identify patients and learn about the timing of the photoimmunotherapy treatment.
Thank you for your comments.
We believe this study will provide supportive information to clinical physicians who are considered which patients for HN-photoimmunotherapy.
Reviewer 3 Report
This is a study about eligibility for photoimmunotherapy in patients with unresectable advanced or recurrent head and neck cancer and changes before and after systemic therapy.
The paper is well written. However, some issues remain.
Please explain what “other” primary site in table 1 means for.
Were all the recurrences at the level of the primary tumor or also in regional nodes?
TNM stages must be reported.
Why were non-SCC included?
How was EGFR-positivity analyzed? If it was not performed, probably the number of eligible patients is lower than reported.
Author Response
[Reviewer 3]
This is a study about eligibility for photoimmunotherapy in patients with unresectable advanced or recurrent head and neck cancer and changes before and after systemic therapy.
The paper is well written. However, some issues remain.
Please explain what “other” primary site in table 1 means for.
The others were squamous cell carcinoma of unknown primary, eyelids, and external auditory canal.
However, in accordance with your later comment, we have re-analyzed the data excluding non-squamous cell carcinoma from this study. Therefore, the relevant others were squamous cell carcinoma of unknown primary, and external auditory canal, and we have included them in the table 2.
Were all the recurrences at the level of the primary tumor or also in regional nodes?
TNM stages must be reported.
The TNM classification was not included in the analysis because it was unnecessary in considering the indications for HN-photoimmunotherapy. Instead, the presence of lesions in the head and neck region, including lymph nodes of neck, was included in the analysis because the head and neck region is considered a possible site for this treatment. Therefore, no data are available, please understand that the TNM classification cannot be added.
Why were non-SCC included?
Because HN-photoimmunotherapy is also treated for non-squamous cell carcinoma in actual clinical practice, it was included in this analysis. In practice, non-squamous cell carcinoma is treated when EGFR expression is confirmed. In this study we don’t have information on EGFR expression. Therefore, it was insufficient and not appropriate to determine the indication including non-squamous cell carcinoma. We have recalculated the data excluding non-squamous cell carcinoma (n=45).
How was EGFR-positivity analyzed? If it was not performed, probably the number of eligible patients is lower than reported.
In response to the above your remark, we re-analyzed the data using only squamous cell carcinoma. It is known that squamous cell carcinomas is highly and strongly expressed EGFR, and cetuximab and this therapy targeted EGFR are treated to head and neck squamous cell carcinomas (HNSCC) without confirmation of EGFR expression. Therefore, this therapy is indicated for the treatment of HNSCC without confirmation of EGFR expression.
Round 2
Reviewer 3 Report
Thanks for improving the mamuscript.
However, not all head and neck tumors have EGFR overexpression. This must be discussed and the lack of EGFR expression must be reported as a limitation of this study.
Author Response
Thank you for your valuable comments.
We do not have information on EGFR expression status in this study. Therefore, we have removed Non-SqCC from the analysis. We stated that at the beginning of the discussion part and added that it is a limitation of this study. (Line 230-232)
Sincerely,